# Affect Regulation Capabilities in Hypermobility Ehlers Danlos Syndrome: Exploring the Associations with Pain Perception and Psychophysical Health

**DOI:** 10.3390/brainsci15020202

**Published:** 2025-02-15

**Authors:** Filippo Camerota, Rachele Mariani, Giada Petronelli, Beatriz Rabissi, Marta Anna Stella Vizzini, Michela Di Trani, Valentina Roselli, Massimo Pasquini, Alessia Renzi, Claudia Celletti

**Affiliations:** 1Physical Medicine and Rehabilitation Division, Umberto I University Hospital, 00161 Rome, Italy; 2Department of Dynamic and Clinical Psychology and Health Studies, Sapienza University of Rome, Via degli Apuli 1, 00185 Rome, Italy; rachele.mariani@uniroma1.it (R.M.); rabissibeatriz@gmail.com (B.R.); marta.vizzini@uniroma1.it (M.A.S.V.); michela.ditrani@uniroma1.it (M.D.T.); alessia.renzi@uniroma1.it (A.R.); 3Physical Medicine and Rehabilitation Division, Sapienza University of Rome, 00185 Rome, Italy; giadapetronelli@gmail.com; 4Department of Neurosciences and Mental Health, Umberto I Policlinic, 00185 Rome, Italy; valentina.roselli@uniroma1.it; 5Department of Human Neurosciences, Sapienza University of Rome, 00185 Rome, Italy; massimo.pasquini@uniroma1.it; 6Department of Life Science, Health, and Health Professions, Link Campus University, 00165 Rome, Italy; clacelletti@gmail.com

**Keywords:** alexithymia, Ehlers–Danlos syndrome, emotion, pain, quality of life, rehabilitation

## Abstract

**Background:** Hypermobility Ehlers–Danlos syndrome (hEDS) is a clinical condition characterized by hypermobility and tissue fragility and is associated with chronic pain. The present study aimed to investigate the associations between affect regulation, pain perception, and psychophysical dimensions as well as alexithymic characteristics in the pathological range. **Methods:** Twenty-five hEDS patients completed a socio-anamnestic questionnaire as well as the Brief Pain Inventory (BPI), the 36-Item Short Form Survey (psychophysical health), the Difficulties in Emotion Regulation Scale (DERS), and the 20-item Toronto Alexithymia Scale (affect regulation). **Results:** Correlational analysis showed several negative significant associations between the SF-36, DERS, and TAS-20. The BPI showed few significant associations with both affect regulation measures. Moreover, a relationship between psychological dimensions and the time since diagnosis emerged. A total of 28% of participants reported TAS-20 scores in the clinical range and 36% reported scores in the borderline area. **Discussion:** Patients with hEDS seem to show high alexithymia levels; pain seems to interfere with the practical aspects of daily life and may reduce an individual’s awareness of their emotional capabilities. The perception of heightened pain has a stronger impact on emotional resources when it interferes with affective life than when it interferes with practical life. Finally, delayed diagnoses of hEDS entail psychological consequences such as alexithymia. **Conclusions:** The present findings highlight the importance of promoting affect regulation capabilities through the implementation of psychological intervention programs for patients suffering from this medical condition.

## 1. Introduction

Ehlers–Danlos syndromes are a heterogenous group of hereditary connective tissue disorders mainly characterized by hypermobility and tissue fragility [1]. The International Classification of EDS and related disorders published in 2017 identifies thirteen clinical subtypes, which are distinguished according to the differential expression of the main phenotypical hallmarks, presence of additional distinctive features, and/or inheritance pattern [2].

A fourteenth type of EDS resembling classic EDS but inherited in an autosomal recessive pattern was described as being associated with biallelic variants in the AEBP1 gene [3].

The most common type of EDS is hypermobile Ehlers–Danlos syndrome (hEDS; formerly EDS type III), which is characterized by joint hypermobility, chronic musculoskeletal pain, skin abnormalities, and easy bruising.

Patients with these symptoms often complain of severe pain [4], which affects quality of life and contributes to the onset or exacerbation of psychopathological symptoms [5]. Various studies report that an excess of emotional symptoms and psychological distress can amplify somatosensory symptoms, enhancing pain perception [6].

Research indicates that affect regulation—namely, the ability to manage the experience and expression of emotions—plays a significant role in understanding the manifestation of symptoms in different medical conditions [7]. Affect dysregulation is considered a risk factor for the development of both physical and mental health diseases as well as for the severity of the disorders [8]. As specifically regards the psychophysical health, the broader literature confirms the presence of higher levels of affect regulation deficits in several somatic and mental conditions compared to healthy controls [9,10,11]. Moreover, variations in emotional awareness and affect regulation capabilities may be connected to the experience of pain in chronic conditions [10,12]. Several constructs within this area have been explored and debated and among them is alexithymia, which has garnered significant theoretical and empirical focus over the past few decades [12]. Alexithymia is a multidimensional construct characterized by challenges in recognizing and articulating emotions, limited imaginative processes, and a tendency toward an externally focused cognitive style [11,13]. More recently, it has been redefined as an emotional dysregulation disorder, encompassing difficulties in using the imagination to manage painful emotions, finding creative solutions to problems, or effectively communicating needs to others to gain social support [9,11]. In general, people showing difficulties in emotional regulation abilities tend to report an increased number of somatic symptoms especially in stressful, negative, or threatening situations and may be considered suffering from hEDS [14,15]. As a consequence of this, individuals with high alexithymia levels and affect dysregulation often experience stronger clinical pain in afflicting situations [16,17,18,19]. A systematic review with meta-analysis revealed that alexithymia was significantly and positively associated with pain intensity and physical interference in chronic pain samples, but when controlled for negative affect-related measures alexithymia was no longer significantly related to pain intensity or interference [12]. Accordingly, the association between alexithymia and pain intensity and interference may be accounted for by negative affect. A recent study revealed greater difficulty in identifying and describing emotions among patients with somatoform pain and depression: while this challenge is partly influenced by elevated levels of depressive feelings, it also persists independently of negative affect [15]. Therefore, Lankes et al. [15] concluded that affect regulation difficulties may worsen the burden of pain and contribute to a poorer outcome. The association between affect regulation and pain is still controversial, producing contrasting results and thus needing to be further explored. To the best of the authors’ knowledge, no previous studies exploring the affect regulation construct in relation to psychophysical health, pain, and disabilities in patients suffering from hEDS have been realized.

The general aim of the present study is to explore the associations between affect regulation, pain perception, and psychophysical dimensions in patients affected by EDS. This is with the broader aim of enriching the knowledge on this clinical population suffering from a rare and difficult-to-diagnose condition.

The hypothesis is that greater difficulties in affect regulation, as evaluated by both high alexithymia levels and affect regulation deficits, will correspond to a greater perception of pain as well as a worse psychophysical health.

## 2. Methods

An observational study, using standardized clinical scales, has been realized in patients with hEDS.

Patients that participated in this study were ambulatorially followed at the rehabilitation program at the “joint hypermobility” outpatient clinic in the Division of Physical Medicine and Rehabilitation of the Umberto I University Hospital (Rome, Italy). Among the total of 150 patients evaluated every year for hereditary connective tissue disorders, in a total period of six months, patients with hEDS who wanted to participate in this study were recruited. Inclusion criteria were a diagnosis of hEDS carried out considering the 2017 International Classification of the Ehlers–Danlos Syndromes and an exclusion of other neurological conditions.

Patients were clinically evaluated and different scales were administered in a total period of 3 months.

The following measures were used to evaluate these patients:*Socio-Demographic Questionnaire*: A socio-demographic questionnaire was specifically designed to collect information concerning age, gender, education level, social status, occupation, and information about the disease.*Difficulties in Emotion Regulation Scale (DERS)* [20,21]. The DERS is a 36-item self-report questionnaire designed to evaluate challenges in emotion regulation, particularly those associated with negative emotions. The scale includes six subscales: Nonacceptance of Negative Emotional Responses (6 items), Difficulties Engaging in Goal-Directed Behavior When Distressed (5 items), Difficulties Controlling Impulsive Behaviors When Distressed (6 items), Lack of Emotional Awareness (6 items), Limited Access to Effective Emotion Regulation Strategies (8 items), and Lack of Emotional Clarity (5 items). Each item is rated on a 5-point scale, ranging from 1 (almost never) to 5 (almost always), and subscale scores are obtained by summing the respective items. The DERS has demonstrated robust psychometric properties. It shows a strong test–retest reliability and high internal consistency across clinical and nonclinical populations [22,23,24].*20-item Toronto Alexithymia Scale (TAS-20)* [25,26]. The TAS-20 is the most widely employed self-report measure to assess alexithymia through 20 items divided into three distinct dimensions: difficulty identifying feelings (DIF), difficulty describing feelings (DDF), and externally oriented thinking (EOT). Participants respond to each item using a 5-point Likert scale ranging from “strongly disagree” (1) to “strongly agree” (5). The scale yields a total score as well as scores for the three subscales, with total scores ranging between 20 and 100. Higher scores indicate higher alexithymia levels. The TAS-20 cut-off scores are as follows: ≤51 no alexithymia, 52–60 borderline alexithymia, and ≥61 alexithymia. The TAS-20 has demonstrated satisfactory psychometric properties, including internal consistency and a strong test–retest reliability.The *36-Item Short Form Survey (SF-36)* is a generic, widely used, and multidimensional tool divided into 8 scales, designed to assess overall health status and capture the impact of a disease on various dimensions of quality of life. The Physical Component Summary (PCS) and Mental Component Summary (MCS) were calculated according to the method developed by Ware et al. [27,28]. The SF-36 measures eight domains of health-related quality of life: Physical Functioning (PF), Role Physical (RP), Bodily Pain (BP), General Health (GH), Vitality (VT), Social Functioning (SF), Role Emotional (RE), and Mental Health (MH). For each domain, a score was calculated and transformed to a value from 0 to 100.The *Brief Pain Inventory (BPI)* [29,30] is a well-recognized tool used to evaluate the severity of pain and its impact on various aspects of a patient’s daily life. The BPI is a patient self-reported 11-point numerical rating scale that measures the severity of pain and the interference of pain on function. There are 4 questions assessing the worst pain, least pain, average pain, and pain right now. The scores range from 0 (no pain) to 10 (pain as severe as you can imagine). The BPI measures both the intensity of pain (sensory dimension) and interference of pain in the patient’s life (reactive dimension), which are activity and affective sub-dimensions (Stanhope 2016). It also queries the patient about pain relief, pain quality, and patient perception of the cause of pain. The BPI is a self-administered measure of the sensory and reactive dimensions of pain: the severity or intensity of the pain and the level of interference it has on various aspects of life.

## 3. Statistical Analyses

The statistical analyses for the present study were executed by means of the Statistical Package for Social Science—24 (SPSS version 24, Armonk, NY, USA). Continuous variables were described as means and standard deviations whereas the discrete variables were reported as percentages and frequencies. Pearson’s correlation analysis was employed to evaluate the associations between affect regulation measures, pain disabilities, and health measures as well as age and time since the diagnosis. As widely shared in the scientific community, we considered r values ranging from 0.400 to 0.599 as moderate/enough and the values ranging from 0.600 to 0.799 as high/strong. A *p* value < 0.05 was considered significant.

## 4. Results

Twenty-eight patients were recruited, but three of them did not fully complete the questionaries for personal reasons (see Figure 1); among the 25 patients studied, only two were male; the mean age at the evaluation was 38.2 ± 17.03.

The socio-anamnestic characteristics of the sample as well as the means and standard deviations of all the measures administered are reported in Table 1 and Table 2, respectively.

Considering the proposed cut-off for the TAS-20, the results showed that 28% of patients had alexithymia and 36% had borderline values. Regarding the SF-36, the Role Physical and Vitality elements showed the worst perception.

Correlational analysis revealed several significant associations between affect regulation measures and the other dimensions investigated, ranging from moderate/enough to high/strong (see Table 3).

Psychophysical health, as evaluated through the SF-36’s dominions, showed several significant associations with both affect regulation measures employed. Specifically, several negative significant associations emerged between the SF-36, DERS, and TAS-20 and difficulties in identifying and describing feelings (see Table 3).

The BPI intensity of pain scale showed only a significant and negative association with the DERS subscale Difficulties Engaging in Goal-Directed Behavior When Distressed. No significances emerged as regards the BPI pain interference with daily life scale (see Table 3).

As specifically regards the anamnestic variables explored, significances emerged for both age and time since the diagnosis. Specifically, age was negatively associated with the SF-36 pain scale (r = −0.530; *p* = 0.026). The time since the diagnosis was negatively associated with the DERS and the TAS-20’s Difficulties in Describing Feelings. It showed a positive association with externally orientated thinking (see Table 3).

## 5. Discussion

Our study aimed to examine the relationship between affect regulation capabilities, pain, and psychophysical health in relation to hEDS patients. To the best of our knowledge, this is the first study to evaluate emotional regulation and specifically alexithymia in this clinical population. We hypothesized that difficulties in affect regulation, indicated by high levels of alexithymia and deficits in emotion regulation strategies, were associated with higher pain perception and worse psychophysical health. We also considered the multifactorial aspects of pain. Our results showed that the role played by psychological strategies, of regulating emotions and being aware of one’s chronic condition, appears to be more complex and intricate than hypothesized. Above all, the role of disease awareness and actual diagnosis in the ability to manage pain and emotional strategies emerges clearly, where the increased time of diagnosis seems to relate to some emotional difficulties. This turns out to be very interesting for understanding the role of pain in these patients, which in this condition has been described as diffuse and related to central sensitization [4].

Upon analyzing the correlations between the pain scales and the alexithymia and affect regulation scales, we observed that the relationship between pain and emotional regulation appeared to be independent. Specifically, the two pain scales, measuring intensity and overall interference, did not show significant correlations with emotional regulation. However, the BPI subscale was revealed to have a significant negative correlation with the emotional regulation awareness strategy. This finding suggests that when pain interferes with the practical aspects of daily life, it may reduce an individual’s awareness of their emotional capabilities.

This insight highlights the intricate interplay between physical pain and emotional regulation, particularly in the context of hEDS, and underscores the importance of addressing both physical and emotional dimensions in therapeutic interventions [31,32].

In a different direction, a correlation emerged between emotional regulation capabilities, specifically the ability to find strategies, and pain perception interfering with affective life. This positive correlation indicates that as pain is increasingly felt in the emotional domain, greater coping abilities are required. This finding suggests that the perception of heightened pain has a stronger impact on emotional resources when it interferes with affective life than when it interferes with practical life.

Notably, the higher the affective perception of pain, the more it is associated with a reduction in emotional awareness. This result aligns with previous research demonstrating that negative emotions amplify pain perception and, conversely, that heightened pain perception can exacerbate negative emotional states [33]. While there is no significant relationship between pain intensity and emotional regulation, similar results were found regarding the relationship between alexithymia and pain. There was no significant correlation between alexithymia factors and pain intensity; however, a positive correlation was observed between pain perception that interferes with affective life and the alexithymia factor “Difficulty Identifying Feelings”. These results suggest that pain perception interfering with emotional life may be a risk factor for psychological health.

The finding of associations between emotional regulation, alexithymia, and quality of life reveals inverse correlations for the factors of Mental Health and Role Emotional [34]. The negative significant correlations show that the higher the quality of emotional life and mental health, the greater the ability to regulate emotions and stay in touch with one’s feelings. This result is consistent with the idea that a good ability to navigate one’s emotional world is possible when it is not impaired by pain perception. In fact, a positive correlation was found between physical function quality and difficulty in being aware of one’s emotional strategies.

Finally, the most intriguing results from this first study examining the relationship between hEDS and psychological variables concern the time of diagnosis. As we know, a critical factor for these patients is the time it takes to diagnose their condition, with the average time it takes to receive a diagnosis being at least 10 years [35]. Our results show that the longer the time since diagnosis, the greater the ability to emotionally regulate and recognize one’s own emotions. This finding underscores the importance of early diagnosis to better understand one’s physical functioning and learn how to manage chronic pain.

However, a counterintuitive result emerged: the time since diagnosis correlates positively with the alexithymia factor of externally oriented thinking. This suggests that knowledge of one’s illness and the development of emotional awareness may trigger defensive strategies, leading to distraction and less constant reflection on one’s inner world. Instead, coping strategies are often more focused on the practical and concrete aspects of life.

The major limitation of this study is the little number of patients. However, we decided to obtain data only from patients directly evaluated; considering the overall prevalence of the disorder being 1 in 5000, our results are representative for a population of 125,000 individuals and therefore can be considered quite significant.

Moreover, socio-cultural factors play an essential role in the way we process and express emotions. It would be interesting to evaluate the differences between ED populations of different socio-cultural contexts. In our study we have observed that in the ED population the levels of alexithymia seem higher than the prevalence of the general Italian population.

Studies on a larger population also related to major clinical data are desirable to confirm and better explain our results.

In conclusion, we have found the presence of alexithymia in the hEDS population like in other conditions of chronic pain (for example, fibromyalgia), and that alexithymia has a relationship with pain. Patients with an impairment in the recognition of their own emotions often find it difficult to describe their situation and to be understood in particular by clinicians. These results also confirm the utility of metaphors [36] as a tool for examining illness experience, which might help clinicians in understanding patient perception and helping them in the management of pain [37].

## Figures and Tables

**Figure 1 brainsci-15-00202-f001:**
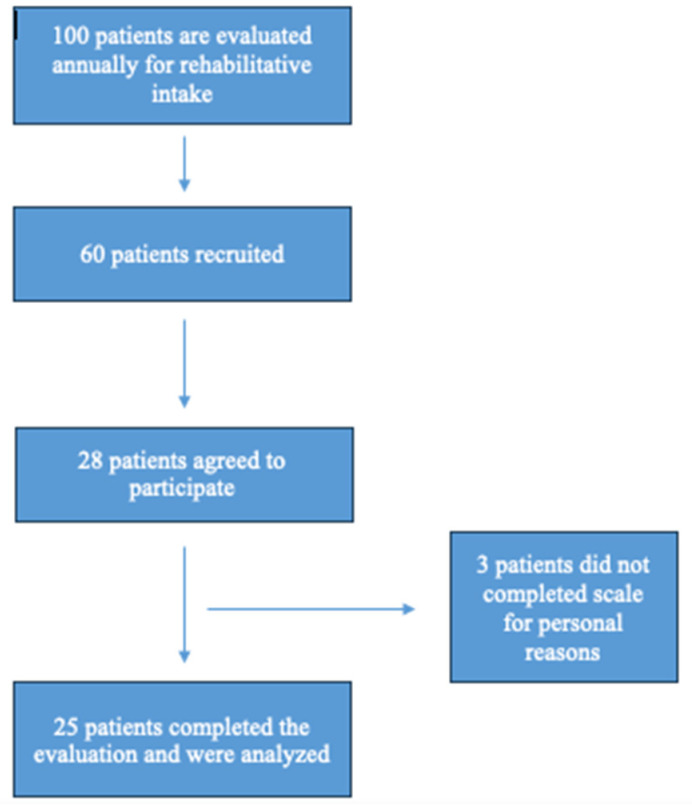
Recruitment diagram.

**Table 1 brainsci-15-00202-t001:** Socio-demographic characteristics.

Socio-Anamnestic Characteristics		
**Variables**	**M/n.**	**SD/%**
Age	38.32	17.01
Time since diagnosis	9.08	6.16
**Employment status**		
Unemployed	4	16
Employed	15	60
Student	6	24
**Educational Qualification**		
Middle School Diploma	3	12
High School Diploma	15	60
Bachelor’s degree	5	20
Postgraduate degree	2	8
**Marital status**		
Single	14	56
Cohabiting	3	12
Married	7	28
Separated	1	4
**Children**		
Yes	14	56
No	11	44

**Table 2 brainsci-15-00202-t002:** Participants’ psychophysical measures investigated.

Variables	M	SD	Minimum Obtained	Maximum Obtained/Possible
**Difficulties in Emotion Regulation Scale**				
Total score	92.24	36.80	40	173/180
Nonacceptance	16.84	8.00	7	30/30
Goals	14.80	5.88	6	25/25
Impulse	14.92	8.14	6	29/30
Awareness	15.12	6.42	6	28/30
Strategies	18.48	9.11	9	38/40
Clarity	12.08	5.49	5	23/25
**20-item Toronto Alexithymia Scale**				
Total score	53.36	12.92	25	76/100
DIF Difficulty Identifying Feelings	15.12	4.72	9	25/25
DDF Difficulty Describing Feelings	21.28	8.03	7	35/35
EOT Externally Oriented Thinking	16.96	4.85	8	26/40
**36-Item Short Form Survey**				
Physical Functioning	56.40	24.85	5	95/100
Role Physical	19.00	27.27	0	100/100
Role Emotional	44.00	38.15	0	100/100
Vitality	33.60	15.31	10	65/100
Mental Health	56.80	19.86	20	92/100
Social Functioning	48.50	22.62	12.5	100/100
Bodily Pain	38.80	16.75	12.5	77.5/100
General Health	26.20	16.85	0	70/100
**Brief Pain Inventory**				
Pain Intensity	5.01	2.13	1	7.75/10
Pain Interference	4.61	2.39	0.71	8.57/10
Activity Pain Interference	4.93	2.40	0.8	9.5/10
Affective Pain Interference	4.65	2.64	0.5	9.5/10

**Table 3 brainsci-15-00202-t003:** Pearson’s r values for the associations between affect regulation and psychophysical dimensions as well as anamnestic variables.

	DERS Total	DERS Nonacceptance	DERS Goals	DERS Impulse	DERS Awareness	DERS Strategies	DERS Clarity	TAS-20 Total	TAS-20 DIF	TAS-20 DDF	TAS-20 EOT
SF-36 Physical Functioning	0.266	0.217	0.153	0.291	0.403 *	0.154	0.142	0.373	0.277	0.243	0.298
SF-36 Role Physical	−0.157	0.265	−0.237	−0.205	−0.293	−0.176	0.177	−0.340	−0.358	−0.180	−0.215
SF-36 Role Emotional	0.010	0.325	−0.674 **	−0.558 **	−0.701 **	−0.462 *	−0.272	−0.719 **	−0.752 **	−0.476 *	−0.559 **
SF-36 Vitality	−0.178	0.001	−0.343	0.007	−0.199	−0.219	−0.241	−0.215	−0.090	−0.406 *	−0.029
SF-36 Mental Health	−0.732 **	−0.687 **	−0.618 **	−0.560 **	−0.460 *	−0.763 **	−0.609 **	−0.680 **	−0.605 **	−0.480 *	−0.340
SF-36 Social Functioning	−0.232	−0.154	−0.221	−0.221	−0.013	−0.267	−0.305	−0.084	−0.118	−0.139	0.109
SF-36 Pain	−0.045	−0.080	−0.062	0.102	0.132	−0.165	−0.149	−0.133	−0.110	−0.114	−0.060
SF-36 General Health	−0.266	−0.051	−0.426 *	0.011	−0.321	−0.271	−0.444 *	−0.153	−0.070	−0.303	0.003
BPI Intensity	−0.168	−0.062	−0.070	−0.225	−0.392	−0.111	0.018	−0.084	0.062	−0.240	−0.093
BPI Interference	0.148	0.187	0.190	0.039	−0.251	0.307	0.242	0.160	0.356	−0.027	−0.137
BPI Activity Interference	−0.059	0.029	0.031	−0.154	−0.439 *	0.124	0.066	−0.012	0.206	−0.140	−0.237
BPI Affective Interference	0.307	0.298	0.322	0.170	−0.041	0.421 *	0.372	0.310	0.448 *	0.133	−0.046
Time Since Diagnosis	−0.530 **	−0.466 *	−0.546 **	−0.618 **	−0.190	−0.435 *	−0.419 *	−0.180	−0.472 *	−0.138	0.453 *

* *p* < 0.05, ** *p* < 0.01. Legend: DERS = Difficulties in Emotion Regulation Scale; TAS-20 = 20-item Toronto Alexithymia Scale; DIF = Difficulties in Identifying Feelings; DDF = Difficulties in Describing Feelings; EOT = Externally Oriented Thinking; SF-36 = 36-Item Short Form Survey; BPI = Brief Pain Inventory.

## Data Availability

The raw data supporting the conclusions of this article will be made available by the authors on request.

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
