# Peer review of "Affect Regulation Capabilities in Hypermobility Ehlers Danlos Syndrome: Exploring the Associations with Pain Perception and Psychophysical Health"

_brainsci, 2025, doi:10.3390/brainsci15020202_

Round 1
Reviewer 1 Report
Comments and Suggestions for Authors
Interesting contribution. Solid methodology.
I wonder if this experience "translates" well into EDS cohorts from other cultures, and maybe this is a point that the Authors could add to the discussion.
I have found the patients of our NY EDS Center to display quite the opposite traits, but I know from direct experience that American patients are very comfortable to discuss emotions with a Medical Professional, while the same cannot be said about Italians (or African American within the US).
Author Response
Interesting contribution. Solid methodology.
Authors: We want to thank you for the time devoted in reviewing the manuscript and for the general positive evaluation. We hope you will find the revised version improved and acceptable for the publication on this valuable journal
I wonder if this experience "translates" well into EDS cohorts from other cultures, and maybe this is a point that the Authors could add to the discussion. I have found the patients of our NY EDS Center to display quite the opposite traits, but I know from direct experience that American patients are very comfortable to discuss emotions with a Medical Professional, while the same cannot be said about Italians (or African American within the US).
Response 1: We thank the reviewer for the possibility to explain something about this aspect. We have added the following sentence in the conclusion:
Moreover, sociocultural factors play an essential role in the way we process and express emotions. It would be interesting to evaluate the differences between ED populations of different sociocultural contexts. In our study we have observed that in the ED population the levels of alexithymia seem higher than the prevalence of the general Italian population.

Reviewer 2 Report
Comments and Suggestions for Authors
1. Recruitment procedures are not clear. In what location/ health service? How many patients were evaluated and recruited? Are these cases just a sample of the total number of patients in the health service? All patients of outpatient facility?
2. In line 162 the author mentions that 3 patients did not complete the questionnaires. Why ? It is necessary to include a flowchart that identifies all stages of study and numbers of patients recruited, included, evaluated, analyzed, as well as, withdrawal cases and their reasons.
3. The study design is unclear. Observational? Cohort? Intervention? It is essential, at the beginning of the methods, to describe the study design and what guidelines guided the design of this protocol.
4. It is not clear what is presented in table 3. Values ​​of which statistical test? p values? media values?
5. In this sense, to corroborate the correlation, it is necessary to include a figure/dispersion graph and curve between the values.

Author Response
Authors: We want to thank the reviewer for the time devoted in reviewing the manuscript and we hope that you will find the revised version improved and acceptable for the publication on this valuable journal.
- Recruitment procedures are not clear. In what location/ health service? How many patients were evaluated and recruited? Are these cases just a sample of the total number of patients in the health service? All patients of outpatient facility?
Response 1: We have modified the paragraph as follow:
Patients that have participate to this study, have been ambulatorial followed for the rehabilitation program at the “joint hypermobility” outpatient clinic in the Division of Physical Medicine and Rehabilitation of the Umberto I University Hospital (Rome, Italy). Among the totally 150 patients evaluated every year for hereditary connective tissue disorders, in a total period of six months, patients with hEDS who wanted to participate to this study has been recruited. Inclusion criteria were diagnosis of hEDS done considering the 2017 International Classification of the Ehlers–Danlos Syndromes and exclusion of other neurological conditions.
2. In line 162 the author mentions that 3 patients did not complete the questionnaires. Why? It is necessary to include a flowchart that identifies all stages of study and numbers of patients recruited, included, evaluated, analyzed, as well as, withdrawal cases and their reasons.
Response 2: Thank you for the observation; we have specified in the text and added a flowchart
3. The study design is unclear. Observational? Cohort? Intervention? It is essential, at the beginning of the methods, to describe the study design and what guidelines guided the design of this protocol.
Response 3: We apologize for the mistake. We have added the following phrase:
An observational study, using clinical scale, has been done in patients with hEDS.
- It is not clear what is presented in table 3. Values ​​of which statistical test? p values? media values?
Response 4 : We thank the reviewer for this comment, which gives us the opportunity to clarify this point. Table 3 presents the correlation analysis conducted to explore the associations between affect regulation variables (TAS-20 and DERS tests) and pain and psychophysical health (SF-30 and BPI), as well as sociodemographic variables (age and time since diagnosis). The table includes r values, and a legend has been added to explain the * values: * p<.05 and ** p<.01. To further clarify this point, the title of Table 3 has been changed to: 'Table 3. Pearson’s r values for the associations between affect regulation dimensions and psycho-physical as well as anamnestic variables.'"
- In this sense, to corroborate the correlation, it is necessary to include a figure/dispersion graph and curve between the values.
Response 5: we appreciate this comment on the importance of make stronger and clear the evaluation of the correlations obtained. In this light we feel that the graphs with only 25 participants and with r values mostly in the area of values going from .400 to .600 and the high number of significances emerged may make not really useful these graphs. At the same time, we believe that adding a general consideration on the interpretation of the values obtained may be useful. We add the following sentences:
Statistical Analysis:
As widely shared in the scientific community, we considered r values ranging from 0.400 to 0.599 as moderate/enough and the values ranging from 0.600 to 0.799 as high/strong.
Results:
Correlational analysis revealed several significant associations between affect regulation measures and the other dimensions investigated ranging from moder-ate/enough to high/strong (see Table 3).
Reviewer 3 Report
Comments and Suggestions for Authors
Dear authors:
In order to improve your manuscript, I would make several comments:
1. TITLE: In my opinion, the title does not correspond to the development of the content of the work. Although the title is focused in the role of alexithymia in the pain perceived by people with EDS, in the abstract you wrote: “The present study aimed to investigate the associations between affect regulation, pain perception and psychophysical dimensions”.
2. ABSTRACT: The conclusion in the abstract are not based on the outcomes of the research but the clinical relevance or the highligths of the study.
3. INTRODUCTION SECTION
You wrote: …Associated to all these symptoms, often patients complain severe pain that influence quality of life and contribute to exhibit personality disorders [5]. Different studies report a considerable excess of emotional symptoms and psychological distress, and somatosensory amplification.
I think the sentence seems to mean that pain contributes to personality disorders and emotional symptoms but that is not clear for me, and, on the contrary, feedback could be possible.
People with hypermobile EDS have a plethora of physical symptoms, such as fatigue, which have a large influence in pain severity. Moreover, there is a great association with biopsychosocial problems as depression or anxiety. New research works support an association to autistic traits in these patients. All those things affect the outcomes of the variables (DERS, alexithymia and pain perception). From my point of view, chronic pain perception in EDS is quite different from other conditions without mental problems associated, as rheumatic arthritis (reference 7).
This topic is not considered in either the introduction section or the discussion section. I encourage you to consider that.
4. RESULTS SECTION
I find this part very interesting.
At the end/foot of every table you should have an explication of the acronyms used and the meaning of the *.
The table 2 could incorporate, in parentheses, the maximum score in each domain of each scale. That could clarify the relevance of the outcomes.
The table 3 is very complicated but I don't know if it is worth separating the variables because it is a good summary of the results.
5. DISCUSSION SECTION
First of all, I would suggest delating this sentence: Our research question focused on understanding how pain is experienced by these patients and its role in the organization of their daily lives. To the best of our knowledge, this is the first study to address this aspect.
This is NOT the first study to address this. I am one of the author publishing this topic in EDS. May be the first one about alexithymia.
I suggest a better specific research strategy.
I add several new references just to show you that this sentence should be rewritten. In addition, I include new ones to justify the comment in “Introduction Section”. Obviously it is not necessary to use all these references.
Finally, I would suggest to discuss the topic before referred and the role of alexithymia in qualitative studies may be reflected in too.
Clark, N.L., Johnson, M., Rangan, A. et al. The biopsychosocial impact of hypermobility spectrum disorders in adults: a scoping review. Rheumatol Int 43, 985–1014 (2023). https://doi.org/10.1007/s00296-023-05298-2
Tage Orenius, Hannu Kautiainen, Marja Louhi, Liisa Montin, Antonio Bulbena, Karl-August Lindgren. Health-Related Quality of Life and Psychological Distress in Patients With Hypermobility Type Ehlers-Danlos Syndrome. SAGE Open. Vol 2022, Issue 4, pp. -10.1177/21582440221091237
Palomo-Toucedo, I.C.; Leon-Larios, F.; Reina-Bueno, M.; Vázquez-Bautista, M.d.C.; Munuera-Martínez, P.V.; Domínguez-Maldonado, G. Psychosocial Influence of Ehlers–Danlos Syndrome in Daily Life of Patients: A Qualitative Study. Int. J. Environ. Res. Public Health 2020, 17, 6425. https://doi.org/10.3390/ijerph17176425
Kennedy, Matthew, Loomba, Katherine, Ghani, Hira and Riley, Bernadette. "The psychological burden associated with Ehlers-Danlos syndromes: a systematic review" Journal of Osteopathic Medicine, vol. 122, no. 8, 2022, pp. 381-392. https://doi.org/10.1515/jom-2021-0267
Rocchetti, M.; Bassotti, A.; Corradi, J.; Damiani, S.; Pasta, G.; Annunziata, S.; Guerrieri, V.; Mosconi, M.; Gentilini, D.; Brondino, N. Is the Pain Just Physical? The Role of Psychological Distress, Quality of Life, and Autistic Traits in Ehlers–Danlos Syndrome, an Internet-Based Survey in Italy. Healthcare 2021, 9, 1472. https://doi.org/10.3390/healthcare9111472
Schubart JR, Mills SE, Francomano CA and Stuckey-Peyrot H (2024) A qualitative study of pain and related symptoms experienced by people with Ehlers-Danlos syndromes. Front. Med. 10:1291189.doi: 10.3389/fmed.2023.1291189
Baeza-Velasco C., Bulbena A., Polanco-Carrasco R., Jaussaud R. Cognitive, Emotional, and Behavioral Considerations for Chronic Pain Management in the Ehlers-Danlos Syndrome Hypermobility-Type: A Narrative Review. Disabil. Rehabil. 2019;41:1110–1118. doi: 10.1080/09638288.2017.1419294. - DOI – PubMed
Berglund B., Pettersson C., Pigg M., Kristiansson P. Self-Reported Quality of Life, Anxiety and Depression in Individuals with Ehlers-Danlos Syndrome (EDS): A Questionnaire Study. BMC Musculoskelet. Disord. 2015;16:89. doi: 10.1186/s12891-015-0549-7. - DOI - PMC – PubMed

Author Response
Dear authors:
In order to improve your manuscript, I would make several comments:
Authors: We want to thank you for the time devoted in reviewing the manuscript and we hope that you will find the revised version improved and acceptable for the publication on this valuable journal
- TITLE: In my opinion, the title does not correspond to the development of the content of the work. Although the title is focused in the role of alexithymia in the pain perceived by people with EDS, in the abstract you wrote: “The present study aimed to investigate the associations between affect regulation, pain perception and psychophysical dimensions”.
Response 1: We want to thank you for this comment. We totally agree with the issue you have highlighted and in accordance with your suggestion we reworded the manuscript’s tile as follows: “Affect regulation capabilities in Hypermobility Ehlers Danlos Syndrome: exploring the association with pain perception and psychophysical health”
- ABSTRACT: The conclusion in the abstract are not based on the outcomes of the research but the clinical relevance or the highligths of the study.
Response 2: We have rewritten the conclusions in accordance with this comment. The revised conclusions are reported below:
Conclusion: The present findings highlight the importance of promoting affect regulation capabilities through the implementation of psychological intervention programs for patients suffering from this medical condition.
- INTRODUCTION SECTION
You wrote: …Associated to all these symptoms, often patients complain severe pain that influence quality of life and contribute to exhibit personality disorders [5]. Different studies report a considerable excess of emotional symptoms and psychological distress, and somatosensory amplification.
I think the sentence seems to mean that pain contributes to personality disorders and emotional symptoms but that is not clear for me, and, on the contrary, feedback could be possible.
Response 3: In response to this comment, these sentences have been reworded to improve clarity for the readers. In general, we strongly believe that pain and the psychological dimensions are mutually influential: experiencing pain may play a significant role in promoting the onset or exacerbation of psychopathological symptoms. At the same time, it is widely recognized that suffering from certain psychopathological symptoms or disorders can represent a risk factor for the development of medical conditions, as well as for their management, as pain perception that influences quality of life.
“Associated with all these symptoms, patients often complain of severe pain [4], which affects quality of life and contributes to the onset or exacerbation of psychopathological symptoms [5]. Various studies report that an excess of emotional symptoms and psychological distress can amplify somatosensory symptoms, enhancing pain perception [6]”
People with hypermobile EDS have a plethora of physical symptoms, such as fatigue, which have a large influence in pain severity. Moreover, there is a great association with biopsychosocial problems as depression or anxiety. New research works support an association to autistic traits in these patients. All those things affect the outcomes of the variables (DERS, alexithymia and pain perception). From my point of view, chronic pain perception in EDS is quite different from other conditions without mental problems associated, as rheumatic arthritis (reference 7).
This topic is not considered in either the introduction section or the discussion section. I encourage you to consider that.
Response: Thank you for the opportunity to better specify this aspect; we have added a sentence in the discussion section.
- RESULTS SECTION
I find this part very interesting.
Authors: We thank you for this positive evaluation.
At the end/foot of every table you should have an explication of the acronyms used and the meaning of the *.
Response 4: In accordance with this suggestion, we have added a legend in Table 3 for the acronyms and the meanings of * and **. Table 2 already includes the full names of the tests within the table.
The table 2 could incorporate, in parentheses, the maximum score in each domain of each scale. That could clarify the relevance of the outcomes.
Response: In accordance with this suggestion we included two columns in Table 2 with minimum score obtained and maximum obtained on possible.
The table 3 is very complicated but I don't know if it is worth separating the variables because it is a good summary of the results.
Response: We agree that Table 3 is very dense; however, it seems to us the best way to present all the results as a whole as also you have highlighted.
- DISCUSSION SECTION
First of all, I would suggest delating this sentence: Our research question focused on understanding how pain is experienced by these patients and its role in the organization of their daily lives. To the best of our knowledge, this is the first study to address this aspect.
This is NOT the first study to address this. I am one of the author publishing this topic in EDS. May be the first one about alexithymia.
I suggest a better specific research strategy.
Response 5: Thank you; we have modified the text considering this observation and focusing the novelty on alexithymia evaluation.
I add several new references just to show you that this sentence should be rewritten. In addition, I include new ones to justify the comment in “Introduction Section”. Obviously it is not necessary to use all these references.
Finally, I would suggest to discuss the topic before referred and the role of alexithymia in qualitative studies may be reflected in too.
Response: Thank you; we have added some useful references recommended by you, and we changed the discussion addressing clearer the role of emotional regulation and alexithymia in relation to pain, as much as a complicated relationship.

Round 2
Reviewer 3 Report
Comments and Suggestions for Authors
Dear authors:
Thanks for considering my suggestions.